# Herbivory and nutrients shape grassland soil seed banks

Anu Eskelinen [1,2,3] ✉, Maria-Theresa Jessen[2,3,4], Hector A. Bahamonde [5], Jonathan D. Bakker [6], Elizabeth T. Borer [7], Maria C. Caldeira [8], W. Stanley Harpole [2,3,9], Meiyu Jia[6,10,11], Luciola S. Lannes [12], Carla Nogueira[8], Harry Olde Venterink [13], Pablo L. Peri[14], Anita J. Porath-Krause[7], Eric W. Seabloom [7], Katie Schroeder[7,15], Pedro M. Tognetti [16,17], Simone-Louise E. Yasui[18], Risto Virtanen [1] & Lauren L. Sullivan[19,20,21,22]

Anthropogenic nutrient enrichment and shifts in herbivory can lead to dramatic changes in the composition and diversity of aboveground plant communities. In turn, this can alter seed banks in the soil, which are cryptic reservoirs of plant diversity. Here, we use data from seven Nutrient Network grassland sites on four continents, encompassing a range of climatic and environmental conditions, to test the joint effects of fertilization and aboveground mammalian herbivory on seed banks and on the similarity between aboveground plant communities and seed banks. We find that fertilization decreases plant species richness and diversity in seed banks, and homogenizes composition between aboveground and seed bank communities. Fertilization increases seed bank abundance especially in the presence of herbivores, while this effect is smaller in the absence of herbivores. Our findings highlight that nutrient enrichment can weaken a diversity maintaining mechanism in grasslands, and that herbivory needs to be considered when assessing nutrient enrichment effects on seed bank abundance.

Global changes, such as nutrient enrichment and shifts in rates of herbivory, can cause long-term changes in plant community dynamics and result in highly interactive effects when multiple drivers influence a single community[1–4]. Anthropogenic nutrient enrichment is a major threat to biodiversity, causing increased competition for light and species loss[5–8]. These effects on composition and richness can be irreversible and may persist for decades even when nutrient supplies are reduced[9–11]. Grazing by mammalian herbivores can counteract these negative impacts of nutrient enrichment by reducing biomass and increasing the amount of light, and thereby maintain diversity[3,8,12]. The benefits to diversity from herbivory can be lost due to extirpation of herbivores or decreased grazing pressure[13–15], which can lead to similarly drastic shifts in plant species composition and reduced diversity[7,8,16,17]. To understand nutrient enrichment and herbivory effects on plant communities, and community and ecosystem resilience in the face of these global change factors, it is critical to examine

the mechanisms that can maintain diversity under global changes and foster or impair community and ecosystem recovery.

Seed banks are a cryptic biodiversity reservoir and a potentially important biodiversity maintaining mechanism. For example, seed banks can increase resilience of plant populations and communities to global changes via the temporal storage effect[18–20], whereby species disperse through time as dormant seeds so they can colonize during benign years[21]. Species have various physiological dormancy mechanisms that inhibit immediate seed germination, which leads to build up of seed banks in the soil and allows species to spread germination over multiple years or prevents them germinating until appropriate environmental cues are met[22]. Thus, seed banks can maintain populations even when they are declining or disappearing from aboveground communities and enhance the likelihood of community recovery from perturbations and inter-annual climatic variability[22–28]. Similarly, seed banks could prove important for

mitigating the effects of nutrient enrichment and loss of herbivory on plant biodiversity.

Nutrient enrichment and herbivory can directly alter seed banks via effects on seed death and germination. For example, some forms of nitrogen can stimulate seed germination in the short-term[29–31] which could deplete seeds from the seed bank in the long-term, if there is no input from aboveground communities. Nutrient enrichment and herbivory can also indirectly affect seed bank composition by altering aboveground community and floral composition, reproductive success and seed rain (i.e., the amount and quality of seeds produced), seed transport and dispersal, and litter accumulation (e.g., 34–37). If the effects of nutrient enrichment and herbivory on aboveground communities are reflected in seed bank communities, we might expect fertilization and absence of herbivores to also decrease seed bank diversity, similarly to aboveground communities. Further, the presence of herbivores could mitigate the negative impacts of nutrient enrichment on seed bank diversity. Seed bank responses may develop if nutrient enrichment and absence of herbivory deplete seeds from seed bank over a long timescale[32] and there is no replacement from aboveground communities[33]. Herbivores also can remove a considerable proportion of preferred flowers or seeds and therefore alter seed rain[34–36], which in turn alters the composition of seeds in seed banks. This indirect alteration of the amount and composition of seed rain could outweigh the otherwise positive effects of herbivores on seed bank richness via increased aboveground community diversity. Herbivores also can transport seeds via epi- and endo-zoochory[37], with potential effects on both aboveground and seed bank communities. Litter accumulation in nutrient enriched and ungrazed conditions[38,39] can trap seeds and prevent seeds from entering the soil, or inhibit germination from seed bank, and the amount of litter can correlate negatively with seed bank abundance[40]. Nutrient enrichment and herbivory can therefore affect seed banks via multiple, sometimes opposing mechanisms, and their effects are also likely to interact. Despite the necessity of considering the joint effects of herbivory and nutrient enrichment when assessing the capacity of seed banks to buffer diversity loss, we are not aware of any such studies.

We collected seed bank data from seven globally distributed Nutrient Network grassland sites with a full-factorial combination of fertilization and herbivore exclusion treatments[41], originating from four continents and encompassing a variety of climatic and edaphic conditions (Fig. 1, Table S1), and examined the individual and joint effects of nutrient enrichment and herbivory on seed banks. We further assessed the extent to which long-term seed banks, i.e., the dormant community of seeds in the soil that have accumulated over the course of years, reflect changes in aboveground communities caused by fertilization and exclusion of herbivory. Earlier studies from Nutrient Network show that fertilization reduces aboveground plant diversity, while herbivory maintains diversity at sites where it

alleviates light limitation[7,42]; however, it is not known whether these effects are reflected in soil seed banks or whether seed banks, by preserving seeds of species previously present in aboveground communities, can maintain plant biodiversity under fertilization and loss of herbivory. We assessed treatment effects on seedling richness (i.e., the number of species in seed banks) and diversity (Shannon diversity and the inverse Simpson), seed bank abundance (i.e., total number of seedlings coming out of the seed banks), and similarity between aboveground and seed bank communities (using Bray-Curtis dissimilarity). We asked the following questions: (1) What are the single and joint effects of fertilization and herbivore exclusion on richness, diversity, abundance, and species composition in seed banks? (2) What are the single and joint effects of fertilization and herbivore exclusion on the similarity between aboveground communities and belowground seed banks? We find that fertilization reduces plant species richness and diversity in seed banks, and increases compositional similarity between aboveground and seed bank communities. Furthermore, we show that fertilization and herbivore exclusion interact to affect seed bank abundance. These results demonstrate that nutrient enrichment can weaken the potential for ecosystem resilience via seed banks and that herbivores can modify seed bank responses to nutrient enrichment.

## Results
### Fertilization and herbivore exclusion effects on species richness, diversity, and abundance of seed banks
We found that fertilization decreased species richness ($\chi^2_{df=1} = 11.86$, $P < 0.001$; Figs. 2a, 3), Shannon diversity ($\chi^2_{df=1} = 16.87$, $P < 0.001$; Fig. 2b), and inverse Simpson diversity ($\chi^2_{df=1} = 10.49$, $P = 0.001$; Fig. 2c) in seed bank communities, while herbivore exclusion had no effect on these response variables (Table S2). In contrast, herbivore exclusion and fertilization interacted to affect total seedling abundance, i.e., the number of seedlings emerging from seed banks ($\chi^2_{df=1} = 6.51$, $P = 0.01$; Fig. 4a). While both herbivore exclusion and fertilization increased total seedling abundance ($\chi^2_{df=1} = 4.56$, $P = 0.03$ and $\chi^2_{df=1} = 10.53$; $P = 0.001$, respectively; Fig. 4a), seedling abundance was greatest in fertilized plots with herbivores present and the joint effect of fertilization and herbivore exclusion was much smaller than the summed effect of individual treatments (i.e., subadditive fertilization × herbivore exclusion interaction; Table S2, Figs. 4a, 5).

When decomposing the total abundance into abundance of functional groups, we found that fertilization increased graminoids ($\chi^2_{df=1} = 11.68$, $P < 0.001$; Table S2, Fig. 4b) and, like total seedling abundance, fertilization and herbivore exclusion interacted subadditively to affect graminoid abundance ($\chi^2_{df=1} = 4.94$, $P = 0.026$; Table S2, Fig. 4b). We found a similar trend in forb abundance (nearly significant fertilization × herbivore exclusion interaction; $\chi^2_{df=1} = 3.68$, $P = 0.055$; Table S2, Fig. 4b); however, forbs were equally abundant in

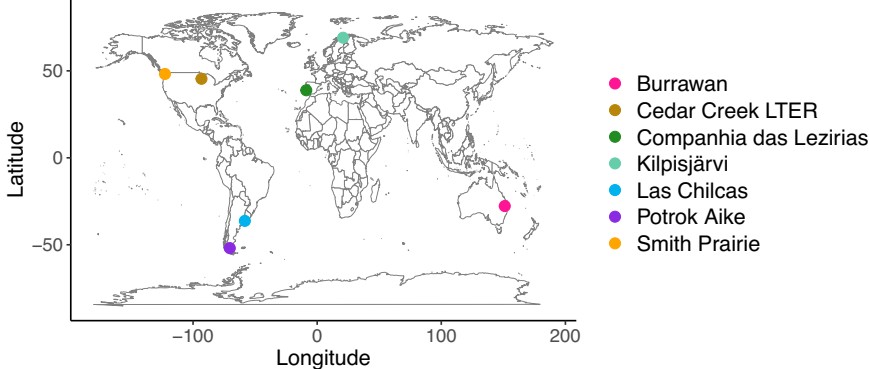

**Fig. 1 | Map showing site locations.** Geographic distribution of the seven experimental sites on four continents from which seed bank samples were collected. For differences in climate and other site-specific information, see Table S1. The map was created using 'maps' package[79] in R Statistical software[75].

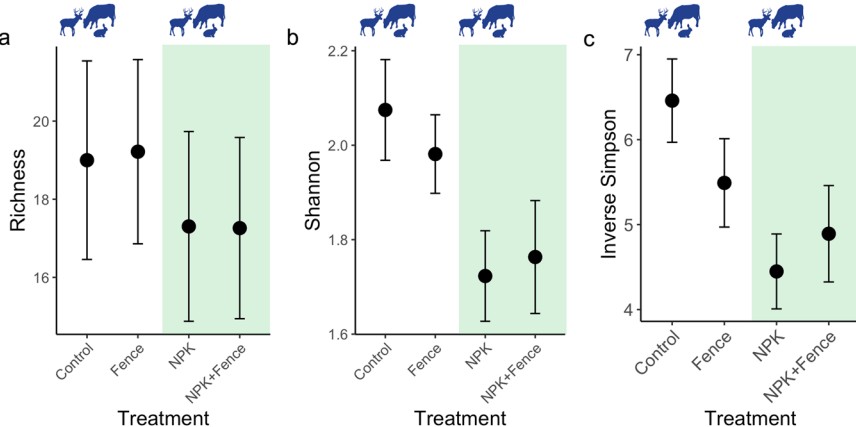

**Fig. 2 | Seedling richness and diversity with respect to herbivore exclusion and fertilization treatments. a** Species richness (number of species), **b** Shannon diversity, and **c** inverse Simpson diversity in seed banks pooled across seven grassland sites in four continents and in different treatment combinations. Points represent data means across sites and error bars represent standard error; *n* = 23 for quadrats from which species richness, Shannon diversity and inverse Simpson diversity were estimated for all treatment combinations. Note that the large variation in richness values results from general between site differences in richness levels, which was taken into account by having site as a random effect in the statistical models. Fence, herbivore exclusion treatment; NPK, fertilization treatment. Green shading indicates fertilized plots and animal symbols indicate plots where herbivores were present. Animal symbols were sourced from PhyloPic (https://www.phylopic.org/).

fenced, fertilized, and fenced plus fertilized plots, and did not peak in fertilized grazed plots.

### Fertilization and herbivore exclusion effects on turnover in seed bank composition, and on similarity between aboveground and seed bank communities

When we assessed how seed bank community composition differed between controls and treatments (exploring spatial treatment differences within a block), we found that treatment altered seed bank composition ($\chi^2_{df=2}$ = 9.64, *P* = 0.008; Fig. 6a). Seed bank communities in herbivore exclusion plots were the least different from controls, while dissimilarity between controls and treatments increased with fertilization, and increased the most with the joint treatment of fertilization and herbivore exclusion (Fig. 6a).

When we assessed dissimilarity between aboveground communities and their corresponding seed banks, we found that fertilization reduced dissimilarity (i.e., increased similarity) between these two communities (significant main effect; $\chi^2_{df=1}$ = 5.39, *P* = 0.02; Fig. 6b, Table S2), but herbivore exclusion had no effect ($\chi^2_{df=1}$ = 1.06, *P* = 0.30; Fig. 6b, Table S2). This provides evidence for homogenization between aboveground and seed bank communities under fertilization (Fig. 6b).

### Fertilization and herbivore exclusion effects on aboveground communities

Aboveground graminoid biomass was significantly higher in fertilized plots ($\chi^2_{df=1}$ = 4.43, *P* = 0.04; Fig. S1), similarly to seed banks, and tended to be higher inside herbivore exclosures ($\chi^2_{df=1}$ = 3.13, *P* = 0.08; Fig. S1, Table S2). However, there was no subadditive fertilization × herbivore exclusion interaction, unlike in seed bank graminoids. Herbivore exclusion and fertilization also interacted to affect forb biomass in aboveground communities: while neither herbivore exclusion nor fertilization alone affected forb biomass, unlike in seed banks, they increased it when applied jointly ($\chi^2_{df=1}$ = 5.27, *P* = 0.02; Fig. S1). Litter mass was positively affected by herbivore exclusion ($\chi^2_{df=1}$ = 5.31, *P* = 0.02; Fig. S1) but not by fertilization (Table S2).

### Discussion

In grasslands on multiple continents and representing a wide range of biotic, climatic, and edaphic site conditions, we found that nutrient enrichment decreased seed bank richness and diversity, altered seed bank community composition, and increased similarity between

aboveground and seed bank communities. Herbivore exclusion had no impact on seed bank richness and diversity, however, it counteracted nutrient-driven increase in seed bank abundance. This effect of herbivore exclusion on total seed bank abundance was due to its distinct effects on graminoids, forbs, and litter. Our results highlight that nutrient enrichment can homogenize composition between aboveground and seed bank communities, reducing the potential for ecosystem resilience via temporal dispersal and the storage effect (i.e., dispersing through time by persisting in the seed bank; 23). Our findings also emphasize the importance of understanding multiple simultaneous global change drivers[4] and the need to consider herbivores when assessing global change effects on seed banks.

Our result that nutrient enrichment decreased seed bank species richness and Shannon diversity in seven grassland sites around the world are in line with results from the few existing single-site studies where fertilization effects on seed banks have been addressed[32,43,44]. Other studies report no change in total seed bank richness in response to fertilization[30,40,45]. Our seven sites represent distinct floras and various climatic and environmental conditions, ranging from a tundra meadow in Northern Europe to a semiarid grassland in Australia (Table S1). Despite this heterogeneity in site conditions, we found a general decline in seed bank richness and diversity, which suggests that negative effects of nutrient enrichment on seed bank diversity are omnipresent, robust to environmental and climatic variation, and species identities. Losses in diversity of seed banks in response to nutrient enrichment reinforce losses of diversity in aboveground communities[7,46] and may help explain the persistent negative effects of fertilization on diversity[9–11].

We found that nutrient enrichment caused divergence in seed bank communities, which is concordant with findings from other studies[33,47]. Added nutrients could affect seed bank composition via multiple mechanisms, for example, by exposing seeds to pathogens[48] that can be facilitated by more nutrient-rich conditions[49,50], or by stimulating more germination that could ultimately exhaust seed banks[29–31]. These impacts can decrease richness and diversity but also change community composition if some species are more susceptible than others due to their seed traits[51]. Further, in our study, nutrient enrichment caused aboveground communities and seed banks to converge, i.e., increased similarity between them. This finding suggests that seed banks were invaded by species that benefitted from fertilization in aboveground communities and reflected dominance changes

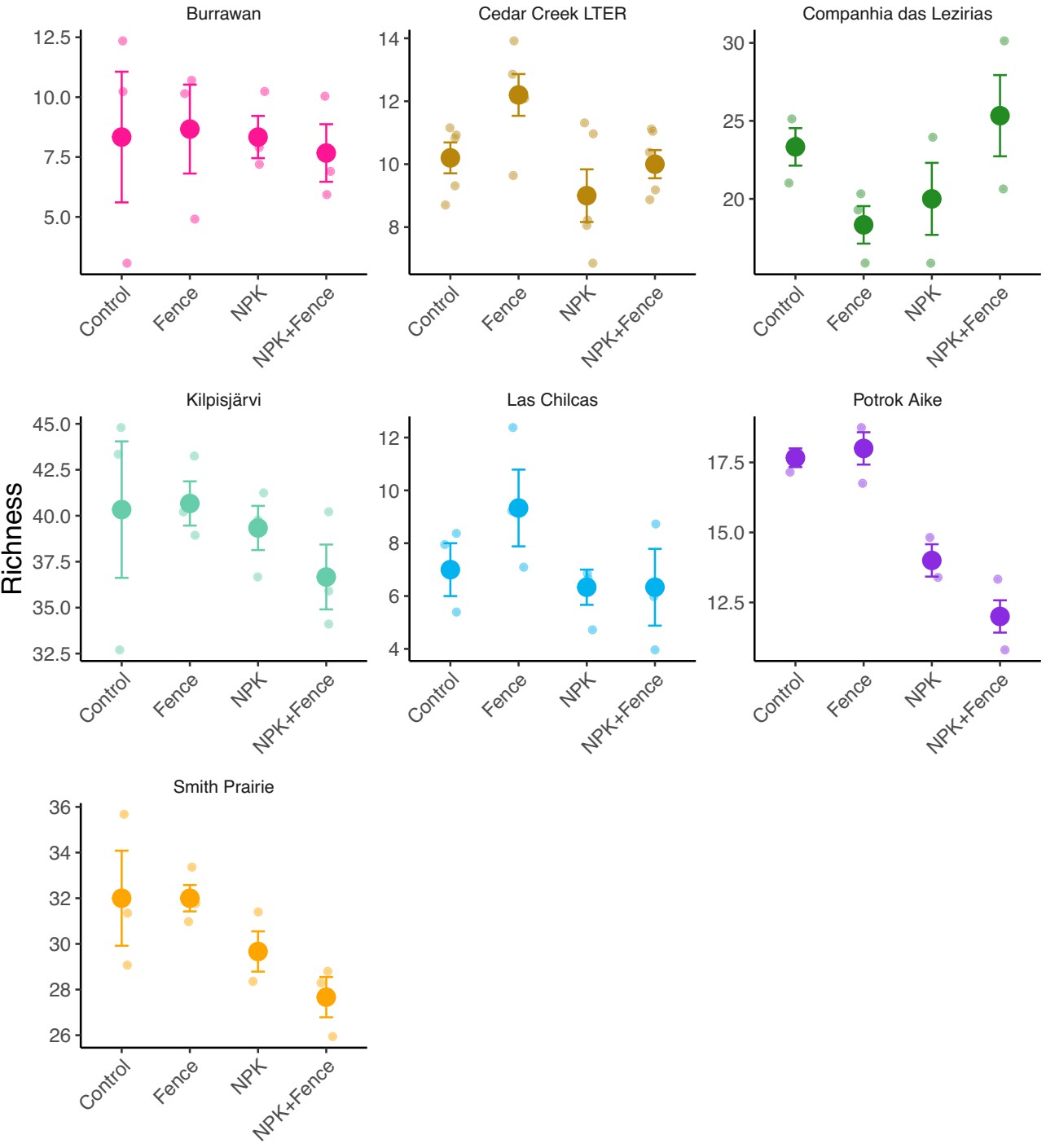

**Fig. 3 | Seedling richness with respect to herbivore exclusion and fertilization treatments at individual sites.** Seed bank species richness (number of species) at the seven individual grassland sites in four continents and in different treatment combinations. Bold points represent data means and error bars represent standard error, with shaded colors behind as individual data points; $n = 3$ for quadrats from which species richness was estimated for all treatment combinations at all sites except for Cedar Creek where $n = 5$ for all treatment combinations. Colors of different sites are as illustrated in the geographical map of the sites (Fig. 1). Fence, herbivore exclusion treatment; NPK, fertilization treatment.

in aboveground communities. This invasion, together with other mechanisms facilitating depletion of species that had accumulated in the seed banks over a long time period, likely resulted in more rapid temporal turnover between above- and belowground communities. These findings from seven grasslands around the world are alarming as they imply that the role of soil seed bank as a temporal diversity storage will be disrupted under eutrophication. As dispersal in time, i.e.,

seed banking, can alter colonization-extinction dynamics in meta-communities, these findings can also have important regional scale implications for biodiversity[21,52].

Herbivore exclusion did not affect seed bank richness and diversity or mitigate fertilization effects. In general, herbivory alone could either increase or decrease seed bank richness[53,54], depending on its intensity and effects on aboveground communities. At our sites, the

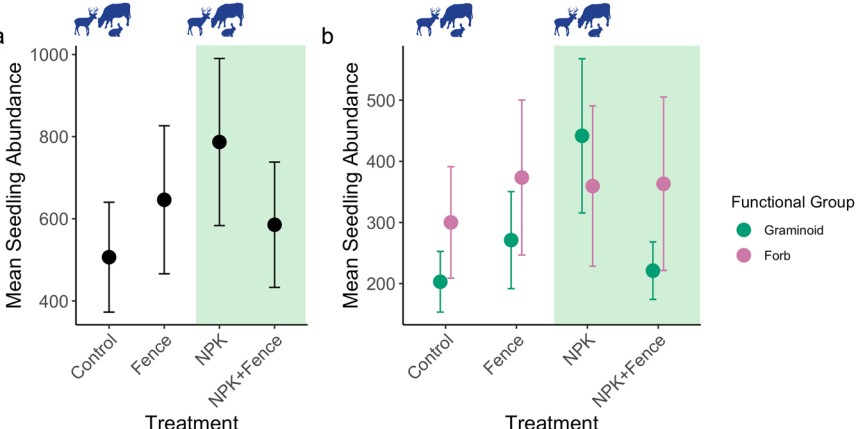

**Fig. 4 | Seed bank abundance with respect to herbivore exclusion and fertilization treatments. a** Total seed bank abundance and **b** the abundance of graminoids and forbs in seed banks pooled across seven grassland sites in four continents and in different treatment combinations. Points represent data means across sites and error bars represent standard error; n = 23 for quadrats from which total seed bank abundance and the abundance of graminoids and forbs were estimated for all treatment combinations. Graminoids include sedges and rushes but consist mostly of grasses. Forbs include legumes but consist mostly of forbs. Note that the large variation in abundance values results from general between site differences in abundance levels, which was taken into account by having site as a random effect in the statistical models. Fence, herbivore exclusion treatment; NPK, fertilization treatment. Green shading indicates fertilized plots and animal symbols indicate plots where herbivores were present. Animal symbols were sourced from PhyloPic (https://www.phylopic.org/).

intensity of herbivory varies[46], and herbivores may not exhibit concordant impact on seed bank diversity, consistent to aboveground community responses[7,46]. However, herbivore exclusion modified fertilization effects on the number of seeds stored in the soil seed bank. Fertilization in the presence of grazers greatly enhanced seed bank abundance while its impact was negligible when combined with exclusion of herbivores. This general pattern of seed bank abundance peaking in fertilized grazed plots coincided with the abundance of graminoids in seed banks, which also exhibited the greatest abundance in fertilized grazed plots, and appeared the main driver of the general seed bank abundance patterns.

Graminoid abundance in seed banks did not fully follow graminoid abundance in aboveground communities, contrasting some other findings (reviewed in ref. 22), as graminoids maintained high biomass also inside fertilized exclosures where graminoids in seed banks diminished. In general, seed bank abundance could be influenced by undecomposed litter as thick litter layer can prevent seeds from entering soil and long-term seed bank[55]. Grass seeds could be especially susceptible to be captured by litter as they are often large, have structures like awns, glumes, and hair. As in other studies[8,39], herbivore exclusion greatly increased litter mass at our grassland sites. It is possible that in fertilized exclosures grasses produced plenty of seeds similarly to fertilized grazed plots, benefitting from fertilization regardless of herbivory; however, in fertilized exclosures seeds were captured by thick litter layer and germinated immediately, failing to enter long-term seed bank that we studied. Granivores could also have been attracted by the thick litter layer[56], consuming graminoid seeds captured by litter, or pathogens could have been thriving in the thick litter layer[57], attacking particularly graminoid seeds. Herbivore effects may therefore be mediated through litter accumulation that isolates long-term seed banks from aboveground communities differently based on functional group. In the longer-term, herbivore exclusion can lead to species-poor seed banks[58], even though it would enhance flower abundance and seed production in the short-term[34,59]. Overall, our findings emphasize that fertilization and herbivore exclusion can exhibit complex interactions on seed bank abundance.

To conclude, our findings are consistent with a recent study that demonstrates the importance of climatic and environmental determinants driving global patterns of seed bank diversity and abundance[60] and extend these observations to suggest that these patterns can be altered by anthropogenic global change drivers. Our

result, that nutrient enrichment engendered greater compositional similarity between aboveground communities and seed banks, suggests that temporal storage mechanisms maintaining populations and biodiversity can be weakened or disrupted by nutrient enrichment. Our findings from seven sites around the world suggest that these effects are common and omnipresent, and may help explain the lack of recovery from fertilization cessation in aboveground communities, found in multiple systems[9–11]. While seed banks may buffer against natural inter-annual climatic variability[26,27], they may have limited capacity to buffer anthropogenic eutrophication. Recovery of these degraded systems to their original condition may therefore require restoration activities, such as supply of seeds. Consistent with other studies that highlight the role of herbivores mediating responses of aboveground communities and ecosystem functioning to global changes[3,7,61,62], our results also underscore their role in modifying seed bank responses to global changes. These findings have implications for assessing global change effects in metacommunity systems, and in conservation, management, and restoration of grassland ecosystems.

## Methods
### Study sites
We collected our seed bank data from seven grassland sites, part of the Nutrient Network collaborative experiment (NutNet, https://nutnet.org), distributed across four continents (Fig. 1). Our sites varied in productivity (168–1570 g m$^{-2}$) and grassland type, ranging from a low-productivity tundra grassland in Finland to a highly productive mesic grassland in Argentina (Table S1). The sites also covered a range of climatic and environmental conditions, with mean annual temperature (MAT) ranging from −3.3 to 18.2 °C and mean annual precipitation (MAP) ranging from 202 to 955 mm (Table S1). The purpose of our study is to assess the generality of seed bank responses to experimental manipulations of nutrients and herbivory in grasslands spanning a wide range of biotic, climatic, and edaphic site conditions.

### Field experiment
The field experiments at all seven sites follow Nutrient Network experimental design with at least three replicate blocks, each consisting of ten 5 × 5 m plots, receiving a combination of different nutrient additions and fencing[41]. To study the effects of nutrient enrichment and herbivore exclusion on soil seed banks, we used the

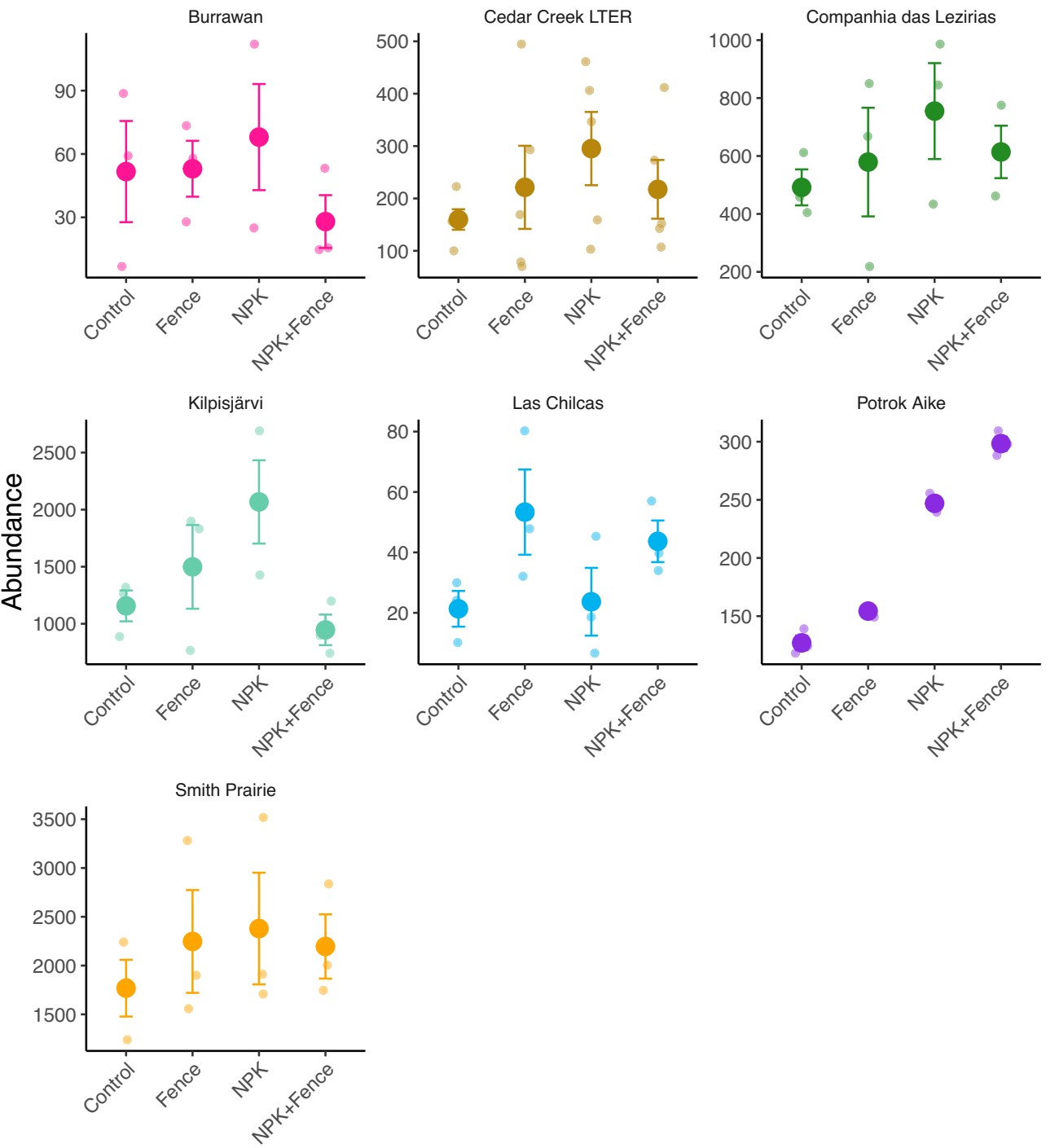

**Fig. 5 | Seed bank abundance with respect to herbivore exclusion and fertilization treatments at individual sites.** Seed abundance (i.e., the number of seedlings that emerged from seed banks) at the seven individual grassland sites in four continents and in different treatment combinations. Bold points represent data means and error bars represent standard error, with shaded colors behind as individual data points; $n = 3$ for quadrats from which species richness was estimated for all treatment combinations at all sites except for Cedar Creek where $n = 5$ for all treatment combinations. Colors of different sites are as illustrated in the geographical map of the sites (Fig. 1). Fence, herbivore exclusion treatment; NPK, fertilization treatment.

full-factorial combination of NPK-fertilization and herbivore exclusion treatments in our seed bank study: (1) control (no NPK-fertilization, no fences), (2) NPK-fertilization (the combined addition of N, P and K, no fences), (3) herbivore exclusion (no NPK-fertilization, fences), and (4) combined treatment of NPK-fertilization and herbivore exclusion (NPK-fertilization, fences). Therefore, each site collected data from at least 12 plots (3 blocks [i.e., replicates] × 4 treatment combinations; see Table S1 for deviations from this number of blocks). At the time of seed bank sampling, the number of years since the initial treatment applications differed between the sites from 2 to 11 years (Table S1). As each of our seven sites is a unique combination of variable biotic, climatic, and edaphic conditions, that also correlate with experimental duration, it was not possible to use this information in our statistical models. However, variation caused by differences in experimental

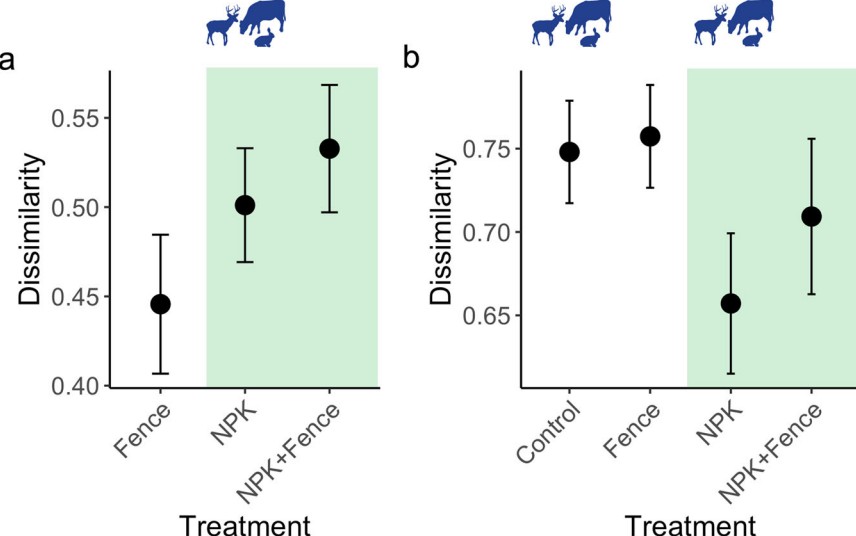

**Fig. 6 | The effects of herbivore exclusion and fertilization on seed bank composition and similarity between seed banks and aboveground communities.** **a** Turnover in seed bank composition with treatment, i.e., Bray-Curtis dissimilarity in seed bank composition between controls and treatments, and **b** Bray-Curtis dissimilarity between aboveground and seed bank communities pooled across seven grassland sites in four continents and in different treatment combinations. Points represent data means across sites and error bars represent standard error; $n = 23$ for quadrats from which turnover in seed bank composition with treatment and Bray-Curtis dissimilarity between aboveground and seed bank communities were estimated for all treatment combinations. Note that the large variation in y-axis values results from general between site differences in abundance levels, which was taken into account by having site as a random effect in the statistical models. Fence, herbivore exclusion treatment; NPK, fertilization treatment. Green shading indicates fertilized plots and animal symbols indicate plots where herbivores were present. Animal symbols were sourced from PhyloPic (https://www.phylopic.org/).

duration is taken into account by using site as a random variable in statistical models (see below).

To implement NPK-fertilization treatment, we applied a mixture of 10 g N m² as time-release urea, 10 g P m² as triple-super phosphate, and 10 g K m² as potassium sulfate to plots receiving NPK-fertilizer treatment at all sites annually. Micronutrient mixture (Fe, S, Mg, Mn, Cu, Zn, B, Mo, Ca; 100 g m²) was applied once at the start of the experiment at each site. Our fertilization treatment was designed to test the effects of nutrient limitation on plant communities and ecosystems in general. As nutrients accumulate to the soil, high experimental nutrient addition rates, such as ours, can also be used to mimic chronic, multidecadal and cumulative nutrient enrichment by nutrient deposition/pollution[63,64]. Further, although our N addition levels are high, they are comparable to N deposition levels in some regions globally[64-67]. Even though N deposition levels also vary between ecosystems, likely being lower in tundra ecosystems, they can be relatively high also in some alpine tundra areas[68,69]. Other nutrients, including P and K, are also present in nutrient deposition, although not in these yearly rates[65,70]. Our grassland sites were natural grasslands that were neither mown nor fertilized for management purposes; therefore our fertilization treatment did not intend to mimic fertilization levels used in grassland management. The time of yearly fertilization varied between the sites (Table S1). To implement herbivore exclusion treatment, we erected a fence 180 cm tall combined with a 1 cm mesh at the lower 90 cm part of the fence, and a 30 cm outward-facing flange stapled to the ground, around plots receiving herbivore exclusion treatment. This treatment aimed at excluding all herbivores more than 50 g except for heavily burrowing and subterranean animals[41]. All sites had natural herbivores, and some sites also had domestic herbivores (Table S1).

### Aboveground plant community sampling
We assessed aboveground plant species composition by visually estimating percentage cover for all plant species in permanently marked 1 × 1 m core plots within each 5 × 5 m plot. We used these data to calculate dissimilarity between aboveground plant communities and seed banks (see Statistical analyses). We also collected litter and live biomass samples from 0.2 × 1 m area next to 1 × 1 m core cover plots in each 5 × 5 m plot. In a laboratory, we sorted live biomass into functional groups (forbs, legumes, graminoids, shrubs, mosses, lichens) that together with the litter were dried at +60 °C to a constant mass (at least 48 h) and weighed to the nearest 0.01 g. We collected these data because litter can affect seeds entering the soil and therefore seed banks, and to allow comparison of functional group abundances in aboveground communities and seed banks. We used composition and biomass data from the same year that the seed bank samples were collected.

### Seed bank sampling and germination
We aimed to characterize the persistent seed bank, i.e., species that were in the soil for more than one year, which represents dormant seeds in the soil. Therefore, we sampled seed bank at each site before the current year's seeds had settled and after the previous year's seeds had germinated[71,72]. At some sites, some sporadic species could have released seeds before majority of seeds were set, contributing current year's seeds to the seed bank; however, our seed bank samples should mostly reflect seeds set during the previous years. Seed bank sampling occurred in 2017-2019, at peak biomass and flowering at each site (Table S1). To assess seed bank composition and richness, we collected four soil squares of 0.1 × 0.1 m at most sites, 5 cm deep, equaling a volume of 2000 cm³, which is higher or corresponding volume compared to other seed bank studies (e.g., 28,38). The samples were collected from 0.2 × 1 m area, spread across the whole area, next to the 1 × 1 m core cover plots and within 5 × 5 m plots (for deviations see Table S1). We chose this depth because most seeds occur in the top 5 cm soil[32,72]. Seed bank samples were brought into the laboratory, and aboveground live vegetation, litter and large rocks were removed. We dried the samples at room temperature and stored them in dry conditions and in closed containers or bags that prevented seeds entering from air until the germination trial began. The storage time of seed bank samples varied depending on the site but was always less than one year.

We used seedling emergence method to determine seed bank composition[71], which is a widely accepted method used in multiple seed bank studies (e.g., [24,26,32,36]). Seed bank samples were transported to a greenhouse where they were germinated in standard warm conditions that varied slightly depending on site (Table S1). We homogenized all soil seed bank cores from each plot, and removed roots, rhizomes, and smaller rocks. The seed bank samples were laid as 1 cm thick layer over 50 × 25 cm trays on top of a 2 cm layer of standard sterile potting soil with perlite (the exact composition varied depending on site). We placed trays in a greenhouse and watered them daily or as needed; trays were checked for emerging seedlings 1–2 times every week and all seedlings were counted and identified into species. We removed seedlings upon identification and replanted those we could not identify as juveniles in separate pots for further identification. At some sites, mosses/algae started growing on the trays and we periodically disturbed the soil surface to prevent moss/crust growth.

Most sites conducted two germination trials separated by a 1–3 month dormancy breaking treatment that was designed to mimic natural conditions at each site (e.g., cold winter or drought, see Table S1 for differences in the type of treatment). We applied this extra treatment to capture seeds that remained dormant during the first trial. Both trials lasted at least for three months and were terminated when no new seedlings had emerged for three weeks. Although we did not check for remaining seeds in the soil samples after the germination trials, the amount of remaining viable seeds should be very low[26,71]. Further, as our seed bank germination trial was identical across treatments, the sampling method should not affect our conclusions concerning the treatment effects.

## Statistical analyses

To determine treatment effects on species richness, and diversity of seed banks, we used linear mixed effects models with Gaussian distributions, and for the treatment effects on seedling abundance (total, graminoids and forbs) in seed banks we used a generalized linear mixed effects model with a negative binomial distribution (distribution determined by AIC, and compared to Gaussian and Poisson) using the 'bbmle' library[73]. All models had random intercepts for blocks nested within sites. Using site as a random factor allowed us to account for site-level variation, i.e., variation caused by environmental, climatic, and other factors that differed between the sites. We defined seedling richness as the number of species (including unique unknowns) that germinated in each plot, plot level diversity as both the Shannon diversity index and the inverse Simpson index as calculated by the 'diversity()' function in the 'vegan' library[74]. We chose these diversity measures because they represent a gradient of the relative importance of rare species (richness) vs. species abundance (inverse Simpson). We defined abundance as the total number of seedlings per plot regardless of species identity. Graminoids included grasses, sedges and rushes but consisted mostly of grasses, while forbs included forbs and legumes but consisted mostly of forbs. All linear mixed effects models were run in R v4.2.2[75] using the 'lmer()' function and generalized linear mixed effects models used the 'glmer.nb()' function, all in the 'lme4' library[76].

We also determined how the treatments impacted the turnover in seed bank communities relative to control plots and between aboveground and seed bank communities. For the aboveground communities, we only examined the vascular components of the community (from the same year as the measured seed bank community) to match the seed bank data. First, we calculated the dissimilarity between the seed bank community in each treatment plot relative to the control plot within a block. We created community matrices for each plot using seedling abundance values and standardized by the total number of seedlings per plot using the 'decostand()' function in the 'vegan' library[74]. Then we compared the standardized community in each treatment plot with its corresponding control plot and calculated the Bray Curtis dissimilarity between these plots using the 'vegdist()' function also in the 'vegan' library. We used Bray-Curtis dissimilarity because it incorporates species abundance whereas presence-absence-based metrics (e.g., Jaccard's) would not capture changes in dominance relationships. We used linear mixed effects models with dissimilarity to control as the response variable, and "treatment" as the predictor variable (which included NPK-fertilizer, fence, and NPK-fertilizer + fence), and block nested within site as a random intercept.

Next, we calculated the turnover between aboveground communities with the seed bank communities. We again created community matrices for the aboveground community (using percent cover values) and the seed bank communities (using abundance values), and standardized both the aboveground composition and the seed bank data by the total abundance and number (respectively) in each plot. In cases where seedlings in the seed bank could only be identified to the genus level at a site, we merged all aboveground species to the genus level at that site as well. Then we calculated the Bray-Curtis dissimilarity between the seed bank composition and the corresponding aboveground composition. We ran linear mixed effects models with the main effects of fertilization and herbivore exclusion treatments and their interaction on the response variable of dissimilarity, with block nested within site as a random intercept. To make sure that our choice of dissimilarity metrics was robust, we tried other abundance-based dissimilarity metrics (i.e., Morisita and Kulczynski) for all dissimilarity analyses above, and found qualitatively similar results for all metrics.

Finally, to assist interpreting the role of aboveground differences in functional group abundance and litter contributing seed bank responses to fertilization and herbivore exclusion, we examined how treatments influenced aboveground biomass (graminoids, forbs, and litter). For this, we used linear mixed effects models to determine how the log of biomass of each functional group was influenced by the interaction of the nutrient addition and herbivore exclosure treatments. We again added block nested within site as a random intercept.

For each statistical model (except where we compared dissimilarity between treatments) we began by examining the interaction between the independent variables of fencing and fertilization, and if no significant interaction was found upon examination with Anova() (see below) we dropped the interaction term and explored the main effects of each treatment. For all tests, significance was determined using the 'Anova()' function with type III sums of squares in the 'car' library[77] which determines significance based on a Wald chi-square statistic (two-tailed). Model fit (of all linear models) was inspected using model diagnostic plots in the 'lme4' library. All marginal and conditional $r^2$ values were calculated with the 'r.squaredGLMM()' function in the 'MuMIn' library[78].

## Reporting summary

Further information on research design is available in the Nature Portfolio Reporting Summary linked to this article.

## Data availability

The data generated in this study have been deposited in the Dryad repository under accession code https://doi.org/10.5061/dryad.w0vt4b8x.

## Code availability

The code generated in this study have been deposited in the Dryad repository under accession code https://doi.org/10.5061/dryad.w0vt4b8x.

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

## Acknowledgements

Coordination of the seed bank study was funded by Academy of Finland (project 297191) to A.E. The seed bank study was conducted within the Nutrient Network (http://www.nutnet.org) experiment, funded at the site-scale by individual researchers. Coordination and data management of aboveground vegetation data from the individual participating sites have been supported by funding to E.T.B. and E.W.S from the National Science Foundation Research Coordination Network (NSF-DEB-1042132) and Long Term Ecological Research (NSF-DEB-1234162 and NSF-DEB-1831944 to Cedar Creek LTER) programs, and the Institute on the Environment (DG-0001-13). Individual sites were supported by PICT-2014-1109 // UBACyT-20020170100191BA // UBACyT-20020190100212BA and visiting researcher fellowship from the Swiss Federal Research Institute WSL to P.M.T., National Natural Science Foundation of China (31170494) and East China University of Technology Research Foundation for Career special project (DHBK2019103) to M.J, and Flexpool grant no. 34600565-11 to M.-T.J. Forest Research Centre is a research unit funded by FCT. We also thank the Minnesota Supercomputer Institute for hosting project data, and the Institute on the Environment for hosting Network meetings. We are grateful to Alvin Barth and Daniel Uhlig for their help with the germination trial, Kilpisjärvi Biological Station that provided facilities during the field work, and Leipzig Botanical Gardens for providing greenhouse facilities. This is Kellogg Biological Station Contribution no. 2352.

## Author contributions

A.E. and L.L.S. conceived the research questions, and designed and coordinated the seed bank study. L.S.S. analyzed the data. E.T.B. and E.W.S. coordinate the Nutrient Network collaboration. A.E., M.-T.J., H.A.B., J.D.B., E.T.B., M.C.C., W.S.H., M.J., L.S.L., C.N., H.O.V., P.L.P., A.P.-K., E.W.S., K.S., P.M.T., S.-L.E.Y., R.V. and L.L.S. contributed data. A.E. wrote the paper and all other authors (M.-T.J., H.A.B., J.D.B., E.T.B., M.C.C., W.S.H., M.J., L.S.L., C.N., H.O.V., P.L.P., A.P.-K., E.W.S., K.S., P.M.T., S.-L.E.Y., R.V. and L.L.S.) contributed to paper writing.

## Competing interests

The authors declare no competing interests.

## Additional information

[1]Ecology and Genetics Unit, University of Oulu, P.O. Box 3000 Oulu, Finland. [2]Department of Physiological Diversity, Helmholtz Centre for Environmental Research - UFZ, Puschstraße 4, 04103 Leipzig, Germany. [3]German Centre for Integrative Biodiversity Research (iDiv), Puschstraße 4, 04103 Leipzig, Germany. [4]Department of Community Ecology, Helmholtz Centre for Environmental Research – UFZ, Halle, Theodor-Lieser-Str. 4, 06120 Halle, Germany. [5]Faculty of Agricultural and Forestry Sciences, National University of La Plata, Av. 60 y 119, La Plata 1900 Buenos Aires, Argentina. [6]School of Environmental and Forest Sciences, University of Washington, Box 354115, Seattle, WA 98195-4115, USA. [7]University of Minnesota, Department of Ecology, Evolution and Behavior, 140 Gortner Laboratory, 1479 Gortner Ave, St Paul, MN 55108, USA. [8]Forest Research Centre, Associate Laboratory TERRA, School of Agriculture, University of Lisbon, Tapada da Ajuda 1349–017 Lisbon, Portugal. [9]Martin Luther University Halle-Wittenberg, am Kirchtor 1, 06108 Halle (Saale), Germany. [10]School of Water Resources & Environmental Engineering, East China University of Technology, Nanchang 330013, China. [11]College of Life Sciences, Beijing Normal University, No. 19 Xinjiekou Wai Street, Beijing City 100875, China. [12]Department of Biology and Animal Sciences, São Paulo State University-UNESP, Ilha Solteira 01049-010, Brazil. [13]Department of Biology, Vrije Universiteit Brussel (VUB), Pleinlaan 2, 1050 Brussels, Belgium. [14]National Institute of Agricultural Research (INTA), Southern Patagonia National University (UNPA), CONICET, Río Gallegos, (CP 9400) Santa Cruz, Argentina. [15]Odum School of Ecology, University of Georgia, Athens, GA 30603, USA. [16]IFEVA, University of Buenos Aires, CONICET, Facultad de Agronomía, Av. San Martin, 4453 C1417DSE Buenos Aires, Argentina. [17]Swiss Federal Institute for Forest, Snow and Landscape Research WSL, Zuercherstrasse 111, 8903 Birmensdorf, Switzerland. [18]Queensland University of Technology, School of Biological and Environmental Sciences, Brisbane QLD 4072, Australia. [19]Division of Biological Sciences, University of Missouri, Columbia, MO 65211, USA. [20]Department of Plant Biology, Michigan State University, East Lansing, MI 48824, USA. [21]W. K. Kellogg Biological Station, Michigan State University, Hickory Corners, MI 49060, USA. [22]Ecology, Evolution and Behavior Program, Michigan State University, East Lansing, MI 48824, USA. ✉e-mail: anu.eskelinen@oulu.fi

