## [Peer Review File · Nature Communications]

Reviewer comments, first round review

Reviewer #1 (Remarks to the Author):

The study of Eskelinen et al. aims to study the separate and joint effects of grazing and nutrient addition on the soil seed banks of grassland ecosystems across the globe. The study is based on a nice, well planned and well replicated experiment which is a part of a large global research network. In general I found the paper well written, and the topic interesting, and there are certainly important knowledge gaps related to the joint effects of grazing and fertilization on seed banks. I have only two major concerns, which are as follows: 1) based on the given information, I am not sure whether this experiment describes real-world scenarios. Usually fertilization is done in order to harvest the fertilized biomass and I am not sure whether this is taken into account. 2) More justification is needed on the seed bank sampling and whether the sampled volume and the number and spatial arrangement of the samples is suitable for describing the density and species composition of the seed bank. Please see my comments related to these points in the detailed review.

Specific comments

L. 59: Here I suggest using the term 'grazing'. Trampling also reduces biomass, and is crucial in forming establishment microsites for subordinate species.

L. 62: here and elsewhere, it would be important to specify the direction of changes. I suppose here not the change is negative but the situation, when an unsuitable grazing intensity is present.

L. 64: The cited references in this sentence refer to climate change, so I suggest specifying 'global changes' as 'global climatic changes' in this sentence. Also this sentence is a bit too general and therefore the knowledge gap highlighted here is not completely clear, and I am unsure whether actually 'very little is known' or probably much is known, but syntheses are lacking.

L. 69: A narrative review about the role of soil seed bank in the resilience of grassland communities under climate change might be interesting here (Kiss et al. 2018, doi: 10.1111/rec.12694)

L. 79-90: this is a nice and important paragraph about potential mechanisms. It might be presented in an even more structured way. E.g. effects of the two drivers on different components, e.g. effect on aboveground floristic composition, effect on reproductive success, effect on establishment microsites and seed traps. An important effect might be mentioned: the role of grazing herbivores in seed dispersal by epi- and endozoochory, which can be probably the most important source of introducing new species into grassland communities.

L. 96: 'grassland sites' (instead of 'grasslands sites')

L. 284-287: The protocol is solid and very well planned. I only have questions whether the applied management regimes represent realistic scenarios that are widely occurring in the studied grassland types. E.g. is it realistic, that on the low productivity tundra grasslands, fertilization is applied? Or in the naturally already very productive grassland?

L. 287-292: Please give information about the management of the experimental units. In reality, fertilization is most often done in order to harvest more biomass. So fertilized grasslands are usually mown, or sometimes grazed (or mown and grazed aftermath). It is a very important point I think to specify whether biomass removal was incorporated into the experiments or not.

L. 301-305: I wonder whether the used NPK treatment is realistic everywhere in the studied gradient. I absolutely see that for such a global network, uniform experimental setup is needed. But I wonder what are the generally applied NPK doses in the different sites and whether the used 10-10-10 g/m² N-P-K reflects the real situation everywhere.

L. 323: In most seed bank studies, long-term persistent seed bank is considered as the component that has been present in the soil for more than five years (not more than one year). So I suggest using the term 'persistent seed bank' and not 'long-term seed bank' here. Persistent seed bank involves both short- and long-term persistent seed bank so I think that 'persistent seed bank' would be an adequate term here.

L. 329-332: could you please clarify the sampled volume per site and how many points per site were the seeds sampled? I know that in long-term experiments, a balance needs to be found between destructive sampling intensity and the keeping the integrity of the whole experiment, so I understand why the sampled soil is limited. But I wonder whether it is enough for representing the communities. One concern is the volume, and the more important concern is the spatial

arrangement of the seed bank sampling. Seed bank often has an aggregated pattern in the soil, so if only a few sampling points are sampled, the full spectrum of the seed bank might be missed.

L. 338: This might be slightly rephrased as I assume that the greenhouses were not situated at the sites.

L. 175: Indeed, there are several studies on the effect of fertilization on soil seed bank. Even a nice long-term experiment by Milberg 1992 *Acta Oecologica* 13: 743-752.

Reviewer #2 (Remarks to the Author):

"Herbivory and nutrients shape grassland soil seed banks" by Anu Eskelinen et al. used data of soil seed bank and aboveground vegetation from seven Nutrient Network grassland sites on four continents to test the effects of fertilization and herbivory on seed banks and the similarity between aboveground plant communities and seed banks. They found that fertilization decreased plant species richness and diversity in seed banks, and homogenized composition between aboveground and seed bank communities. Seed banks can increase resilience of plant populations and communities to global changes. The results of this study are very interesting and important for understanding the effect of global change on plant communities. I have following comments.

1. In Introduction, the second paragraph is on the role of seed bank to increase resilience of plant populations and communities to global changes. However, this paragraph is very short and need to be expanded to include physiological mechanism.

2. In Methods, the month of NPK-fertilizer should be provided. In addition, it should be clear how the plots were fertilized (once or twice per year).

3. It should be clear how long the seed bank sample was stored before germination trial. It is important because storage can significantly affect seed viability in seed bank.

4. Some seeds in soil can be dormant for long time, therefore the germination trays should be checked for seeds after germination trial.

5. Line 74-77, needs references for these sentences.

6. Line 148-156, these should be another section, because this is only on aboveground community.

7. Line 159-160, this sentence is not needed.

8. Line 632 and 633, "consists" should be "consist".

Reviewer #3 (Remarks to the Author):

Nature Comms review

Overall, I thought that this was an interesting paper, although not groundbreaking. In many ways, the results are a logical consequence of the treatment effects on aboveground communities, that are already published. After all, the seedbank must reflect the aboveground community.

That said, nutrient network is a great example of a distributed platform, and so has the advantage of being able to generalise findings over multiple sites. This makes contributions from the network more valuable than many single-site experiments.

Major issues

I think this paper needs to be more closely linked to the aboveground community findings. It's almost written as if no-one's ever bothered to look at the impacts of nutrient additions and herbivore exclusions on the aboveground community, whereas that seems to me to be the obvious starting point for the paper. In other words, given what you have already found aboveground, what's now happening with the seedbank? Is it just following suit, or are there some surprises? I also don't think it's reasonable to expect your readers to know your entire output of work – hence the need to restate the main results here in the introduction, and not force your reader to go off and read further papers in order to properly understand this one. The hypotheses could also be

framed around the aboveground findings – this would make the whole thing much easier to understand.

Second, I think the figure showing how the different sites respond (Figure S2) should be in the main paper. Otherwise, we really don't know how consistent site responses are (given that you fit site as a random effect and don't look for treatment*site interactions). It's good to see that most sites do indeed respond in a very similar way and this strengthens your paper for me. You could probably leave Figure S3 in the SI, as this result is probably less important than the species richness result.

Minor issues

In general, the paper is not particularly easy to read. In particular, the discussion is way too long, given the brevity of the rest of the paper and it ends with a long wordy conclusion that just recapitulates the abstract. I would suggest reducing it from its current six paragraphs (+conclusion) to three (+conclusion if this is what the journal demands). I think the authors are perfectly capable of judging which of the less important things they would like to remove without specific input from me.

Responses to reviewer comments

REVIEWER COMMENTS

Reviewer #1 (Remarks to the Author):

1. The study of Eskelinen et al. aims to study the separate and joint effects of grazing and nutrient addition on the soil seed banks of grassland ecosystems across the globe. The study is based on a nice, well planned and well replicated experiment which is a part of a large global research network. In general I found the paper well written, and the topic interesting, and there are certainly important knowledge gaps related to the joint effects of grazing and fertilization on seed banks. I have only two major concerns, which are as follows: 1) based on the given information, I am not sure whether this experiment describes real-world scenarios. Usually fertilization is done in order to harvest the fertilized biomass and I am not sure whether this is taken into account. 2) More justification is needed on the seed bank sampling and whether the sampled volume and the number and spatial arrangement of the samples is suitable for describing the density and species composition of the seed bank. Please see my comments related to these points in the detailed review.

Our response: We thank the Reviewer for these constructive words. We are grateful for a thorough review and comments/suggestions that greatly helped us to improve our manuscript. In brief, we have now described and justified our fertilization treatment in detail (please see our detailed responses to comments no 8 and 10 below). We have also described and justified our seed bank sampling and added lots of details to the manuscript (please see our detailed response to comment no 12 below). We have also made numerous other changes to the manuscript based on the suggestions from the Reviewer; please see our detailed responses/explanations below.

Specific comments

2. L. 59: Here I suggest using the term 'grazing'. Trampling also reduces biomass, and is crucial in forming establishment microsites for subordinate species.

Our response: We now use here "grazing by mammalian herbivores", please see line 63.

3. L. 62: here and elsewhere, it would be important to specify the direction of changes. I suppose here not the change is negative but the situation, when an unsuitable grazing intensity is present.

Our response: Thank you for this comment. In principle, both intense and low grazing pressure could cause diversity to decline, although here we refer to studies that report decreased grazing pressure and its effects on plant communities. We modified the text, please see line 66.

4. L. 64: The cited references in this sentence refer to climate change, so I suggest specifying 'global changes' as 'global climatic changes' in this sentence. Also this sentence is a bit too general and therefore the knowledge gap highlighted here is not completely clear, and I am unsure whether actually 'very little is known' or probably much is known, but syntheses are lacking.

Our response: Thank you for this comment that helped us understanding that this last sentence was not capturing what we wanted to say. We have now modified the end of this paragraph, please see lines 68-71.

5. L. 69: A narrative review about the role of soil seed bank in the resilience of grassland communities under climate change might be interesting here (Kiss et al. 2018, doi: 10.1111/rec.12694)

Our response: Thank you for this excellent suggestion; we have now added this reference to the manuscript (line 82).

6. L. 79-90: this is a nice and important paragraph about potential mechanisms. It might be presented in an even more structured way. E.g. effects of the two drivers on different components, e.g. effect on aboveground floristic composition, effect on reproductive success, effect on establishment microsites and seed traps. An important effect might be mentioned: the role of grazing herbivores in seed dispersal by epi- and endozoochory, which can be probably the most important source of introducing new species into grassland communities.

Our response: Thank you for these great suggestions. We added epi- and endozoochory as mechanisms by which herbivores can affect both above- and belowground communities, see lines 101-102. We have also restructured the paragraph, highlighting the factors the Reviewer pointed, thought kept our original classification in direct and indirect effects. Please see lines 84-105.

7. L. 96: 'grassland sites' (instead of 'grasslands sites')

Our response: Thank you, this is now corrected.

8. L. 284-287: The protocol is solid and very well planned. I only have questions whether the applied management regimes represent realistic scenarios that are widely occurring in the studied grassland types. E.g. is it realistic, that on the low productivity tundra grasslands, fertilization is applied? Or in the naturally already very productive grassland?

Our response: We appreciate this question and agree that it is important to clarify this matter. Our nutrient addition manipulation was intended to test both theory of nutrient limitation and also to mimic nutrient enrichment as a global change factor (via airborne nutrient deposition). We have now stated this more clearly in the description of our study, please see lines 304-307.

We fully agree with the reviewer that it would not make sense to manage a tundra grassland by fertilization. Our grassland sites around the world are, however, natural grasslands and fertilization is not used as a management type at any sites (we added information about this to lines 311-312). Natural productivity of the grasslands varies from very low-productivity tundra grassland in Finland to highly productive mesic grassland in Argentina, showing a total biomass range from 168 to 1570 g m⁻² (see lines 275-277). As far as we know, using fertilization as a management type in grasslands has been decreasing globally, as fertilizing is too expensive relative to the gains to make it worth the price in most places. Mimicking fertilization as a management type was therefore not our purpose. We have now clearly stated this, please see lines 313-314.

Our questions and interests rather relate to nutrient limitation and the effects of nutrient enrichment via anthropogenic sources, and how important these are affecting aboveground communities and their corresponding seed banks across grasslands globally. Although our sites considerably differ in their original productivity and nutrient concentrations, we still find very similar responses. We believe that it is valuable information that even highly productive and nutrient-rich sites still show similar response to nutrient enrichment.

Please also see our response to comment no 10.

9. L. 287-292: Please give information about the management of the experimental units. In reality, fertilization is most often done in order to harvest more biomass. So fertilized grasslands are usually mown, or sometimes grazed (or mown and grazed aftermath). It is a very important point I think to specify whether biomass removal was incorporated into the experiments or not.

Our response: Thank you for making this point; we agree that this is important information. None of the Nutrient Network sites that participated this seed bank study mow their sites, i.e., biomass is not collected. We added this information to the Methods, please see lines 311-312.

10. L. 301-305: I wonder whether the used NPK treatment is realistic everywhere in the studied gradient. I absolutely see that for such a global network, uniform experimental setup is needed. But I wonder what are the generally applied NPK doses in the different sites and whether the used 10-10-10 g/m² N-P-K reflects the real situation everywhere.

Our response: Our fertilization treatment was designed to test the effects of nutrient limitation on plant communities and ecosystems in general. Since our goal is to overcome nutrient limitation, rates are set to be high enough to do this while still not harming plants. As nutrients can accumulate to the soil, high experimental nutrient addition rates, such as ours, can also be used to mimic chronic, multidecadal and cumulative nutrient enrichment by nutrient deposition/pollution (Dupré et al. 2010, Borer & Stevens 2022). Further, although our N addition levels are in general high, they are comparable to N deposition rates in some highly impacted regions globally (Schlesinger et al. 1982, Fenn et al. 2003, Ackerman et al. 2018, Galloway et al. 2008, Borer & Stevens 2022), and landscape-level inputs can be larger (Sutton et al. 2007, Galloway et al. 2008). Other nutrients, including P and K, are also present in nutrient deposition (Schlesinger et al. 1982, Newman 1995). Therefore, the fertilization levels at some of our sites can be considered high (e.g., in tundra), while they should correspond to nutrient deposition levels in some systems (e.g., levels in some parts of Europe, some areas in the US), and can well mimic cumulative effects of nutrient deposition. On the other hand, even some alpine tundra areas (e.g., those in Central Europe) can experience relatively high levels of nutrient deposition (Galloway et al. 2004, Dentener et al. 2006), and our nutrient manipulation can give valuable information about nutrient enrichment effects in such highly affected alpine tundra areas. We have now added information about nutrient deposition levels to Methods, please see lines 305-311.

As the reviewer acknowledges, since this is a global network, our treatment needs to be uniform at all sites. If the treatment levels varied between sites, for example, depending on site specific nutrient deposition levels, we would essentially have as many fertilization treatments as there are sites. It would be impossible to compare site responses and tell whether differences between site responses to fertilization are caused by differences in fertilization dosage or by other site-specific factors. Therefore, fertilization levels need to be uniform across sites, regardless of the original site N deposition levels or nutrient-richness of the site. Furthermore, although our sites considerably differ in their original productivity and nutrient concentrations, we still find surprisingly similar responses. We believe it is valuable information that even highly productive and nutrient-rich sites show similar response to nutrient enrichment than originally relatively nutrient-poor sites. This should make our findings generalizable across many systems.

Please also see our response to comment no 8.

11. L. 323: In most seed bank studies, long-term persistent seed bank is considered as the component that has been present in the soil for more than five years (not more than one year). So I suggest using the term 'persistent seed bank' and not 'long-term seed bank' here. Persistent seed bank involves both short- and long-term persistent seed bank so I think that 'persistent seed bank' would be an adequate term here.

Our response: We agree and have changed the terminology as suggested (line 333).

12. L. 329-332: could you please clarify the sampled volume per site and how many points per site were the seeds sampled? I know that in long-term experiments, a balance needs to be found between destructive sampling intensity and the keeping the integrity of the whole experiment, so I understand why the sampled soil is limited. But I wonder whether it is enough for representing the communities. One concern is the volume, and the more important concern is the spatial arrangement of the seed bank sampling. Seed bank often has an aggregated pattern in the soil, so if only a few sampling points are sampled, the full spectrum of the seed bank might be missed.

Our response: We apologize for the missing information. We have now added information about the sampled volume and number of points sampled, please see lines 341-343.

Briefly, we collected four 10 x 10 cm soil corers (5 cm deep; this is where most of the seeds occur) that were distributed across 20 x 100 cm area. The total volume of the sampled area is therefore 2000 cm³, which is more than in many seed bank studies (e.g., 1178 cm³ in Basto et al. 2015 Nat Comm, 1250 cm³ in DeMalach et al. 2020 J Ecol). We agree with the reviewer that usually seed bank samples are collected from relatively small area, and this is because germinating seeds out of the soil samples is a lot of work and takes a lot of greenhouse space. Soil samples were spread over a tray as a thin layer and the more soil there is, the more area one sample takes (i.e., the more trays). However, as pointed earlier, our sampling was conducted according to a standard protocol, with higher or comparable volume of soil than in other studies.

In our experiment, the purpose was to compare seed banks in different treatments and test whether fertilization, herbivore exclusion and their interaction changed seed bank communities. As our sampling was identical in different treatments, our results, i.e., seed bank responses to the treatments, should be unaffected by the area/volume sampled. Therefore, the sampled area/volume should not affect conclusions from our study (please see lines 370-372).

13. L. 338: This might be slightly rephrased as I assume that the greenhouses were not situated at the sites.

Our response: Thank you for noting this, we deleted "At each site" and slightly modified the sentence (see lines 352-353).

14. L. 175: Indeed, there are several studies on the effect of fertilization on soil seed bank. Even a nice long-term experiment by Milberg 1992 Acta Oecologica 13: 743-752.

Our response: Thank you for pointing us this excellent paper. We now refer to it in our manuscript (see line 195).

Reviewer #2 (Remarks to the Author):

15. "Herbivory and nutrients shape grassland soil seed banks" by Anu Eskelinen et al. used data of soil seed bank and aboveground vegetation from seven Nutrient Network grassland sites on four continents to test the effects of fertilization and herbivory on seed banks and the similarity between aboveground plant communities and seed banks. They found that fertilization decreased plant species richness and diversity in seed banks, and homogenized composition between aboveground and seed bank communities. Seed banks can increase resilience of plant populations and communities to global changes. The results of this study are very interesting and important for understanding the effect of global change on plant communities. I have following comments.

Our response: We thank the reviewer for appreciating our study. We are grateful for these comments and have done our best to improve the manuscript based on them. Please see our detailed responses to specific comments below.

16. In Introduction, the second paragraph is on the role of seed bank to increase resilience of plant populations and communities to global changes. However, this paragraph is very short and need to be expanded to include physiological mechanism.

Our response: Thank you; we have now added some text about dormancy-related mechanisms to this paragraph (see lines 76-83). However, as we do not specifically study physiological mechanisms, we wish to keep this part short.

17. In Methods, the month of NPK-fertilizer should be provided. In addition, it should be clear how the plots were fertilized (once or twice per year).

Our response: We have now added information about the month when fertilization is usually done at each site to the Supplementary Table S1. However, since our sites are located at different sides of the globe, both in the southern and northern hemisphere, the month of the fertilization is not necessarily very informative about at which stage of the growing season fertilization was done. We therefore also added information about the time of growing season (early, mid, later growing season or peak biomass) when the fertilization was done (Table S1). Further, we also added information about how many times fertilization is done at different sites. Most sites fertilize only once while some split the annual dose in two or three parts; Table S1.

18. It should be clear how long the seed bank sample was stored before germination trial. It is important because storage can significantly affect seed viability in seed bank.

Our response: Seed bank samples were stored different times at different sites, but was always clearly less than one year. We have now added this information to the Methods, see lines 349-350.

19. Some seeds in soil can be dormant for long time, therefore the germination trays should be checked for seeds after germination trial.

Our response: We appreciate this comment and understand that this issue is important to discuss. In our study, we used the “seedling emergence method” (Thompson et al. 1997), which does not include checking of seeds after germination trial. This method is widely accepted and used in multiple seed bank studies (e.g., Basto et al. 2015 Nat Comm, Vandvik et al. 2016 Oikos, LaForgia et al. 2018 Ecology, DeMalach et al. 2020 JEcol, Eskelinen et al. 2021 Ecology, Auffret et al. 2023 JEcol). None of these studies check for remaining seeds and their viability after germination trial. Some studies do not even report their method of seed bank determination (Plue et al. 2021 GEB, Yang et al. 2021 Nat Comm). The percentage of viable seeds remaining in soil samples after 3-12 months of germination should be very low (Thomson et al. 1997 and references therein, DeMalach et al. 2020 JEcol). For example, DeMalach et al. 2020 found that the number of seeds found from seed bank samples after germination trail was <1% of the total number of emerged seedlings, and probably only a fraction of these seeds were viable (DeMalach et al. 2020 did not check for viability). Our own experience suggests that relatively little germination occurs after the first germination trial, even after a germination triggering treatment between the trials (Eskelinen et al. 2021, results from this study). We have now added a justification to our method to the Methods, please see lines 351-352 and 369-370.

Importantly, in our experiment, the purpose was to test whether fertilization, herbivore exclusion and their interaction changed seed bank communities. i.e., compare seed banks in different

treatments. As our seed bank sampling/identification method was identical in different treatments, our results, i.e., seed bank responses to the treatments, should be unaffected by any error that the sampling method could cause. So, the sampling method should not affect conclusions from our study; please see lines 370-372.

20. Line 74-77, needs references for these sentences.

Our response: This part of the text has been changed based on comments from Reviewer 1. We also added references; please see lines 85-86 and the whole paragraph.

21. Line 148-156, these should be another section, because this is only on aboveground community.

Our response: Thanks, we added a separate subheading for aboveground community responses; please see line 168.

22. Line 159-160, this sentence is not needed.

Our response: We deleted this sentence as suggested.

23. Line 632 and 633, “consists” should be “consist”.

Our response: Corrected.

Reviewer #3 (Remarks to the Author):

Nature Comms review

24. Overall, I thought that this was an interesting paper, although not groundbreaking. In many ways, the results are a logical consequence of the treatment effects on aboveground communities, that are already published. After all, the seedbank must reflect the aboveground community.

That said, nutrient network is a great example of a distributed platform, and so has the advantage of being able to generalise findings over multiple sites. This makes contributions from the network more valuable than many single-site experiments.

Our response: We thank for these helpful comments and are grateful that the reviewer appreciates our attempt to generalize fertilization and herbivore exclusion effects on seed banks across grasslands in different parts of the world and representing different climatic and edaphic conditions. We would also like to note that while fertilization effects on seed banks have been often examined, these previous studies do not consider how herbivores might interact with fertilization effects, even though we know that these two factors interact to affect aboveground communities. Indeed, while seed banks in general should reflect aboveground communities, as the reviewer notes, seed bank richness and composition have been shown to be decoupled from variation in aboveground communities caused by environmental and interannual climatic variation (DeMalach et al. 2020, Plue et al. 2021), acting as a source of new recruits and promoting recovery. So far it has been unclear whether they can provide similar resistance to the effects caused by fertilization and loss of herbivory and based on our results they cannot.

Major issues

25. I think this paper needs to be more closely linked to the aboveground community findings. It's almost written as if no-one's ever bothered to look at the impacts of nutrient additions and herbivore exclusions on the aboveground community, whereas that seems to me to be the obvious

starting point for the paper. In other words, given what you have already found aboveground, what's now happening with the seedbank? Is it just following suit, or are there some surprises? I also don't think it's reasonable to expect your readers to know your entire output of work – hence the need to restate the main results here in the introduction, and not force your reader to go off and read further papers in order to properly understand this one. The hypotheses could also be framed around the aboveground findings – this would make the whole thing much easier to understand.

Our response: This is a great comment, and we fully agree that incorporating aboveground community responses to fertilization and herbivore exclusion in the same experimental system as a background information is important. We have now added some text on lines 117-122.

26. Second, I think the figure showing how the different sites respond (Figure S2) should be in the main paper. Otherwise, we really don't know how consistent site responses are (given that you fit site as a random effect and don't look for treatment*site interactions). It's good to see that most sites do indeed respond in a very similar way and this strengthens your paper for me. You could probably leave Figure S3 in the SI, as this result is probably less important than the species richness result.

Our response: We agree and have moved Figures S2 and S3 to the main manuscript (now Figs 3 and 5).

Minor issues

27. In general, the paper is not particularly easy to read. In particular, the discussion is way too long, given the brevity of the rest of the paper and it ends with a long wordy conclusion that just recapitulates the abstract. I would suggest reducing it from its current six paragraphs (+conclusion) to three (+conclusion if this is what the journal demands). I think the authors are perfectly capable of judging which of the less important things they would like to remove without specific input from me.

Our response: Thank you for this comment. We have now cut the length of the Discussion by approximately one page. Currently Introduction, after the additions that the Reviewers 2 and 3 asked, is about 3 pages (920 words). The discussion is currently 3.5 pages (1120 words). The length of the Discussion should therefore be well proportional to the length of the Introduction and is well in line or much below the length of some recent ecological papers published in Nature Communications (see, for example, Ghilardi et al. 2023 Nat Comm 14:985, Kambach et al. Nat Comm 14:712). Cutting the length of the Discussion by almost half, as the reviewer suggests, would make it much shorter than Introduction (i.e., disproportionately short). In our opinion, it would also jeopardize discussing essential main results to the extent that they need to be discussed for them to be understandable for readers. Alternatively, we would completely need to disregard discussing some of the main results, omitting important information and affecting the quality of the manuscript. Therefore, we are hesitant to make such drastic changes to the manuscript that the reviewer requests without editorial consultation. We would be grateful for editorial guidance in this matter.

References not appearing in the manuscript:

Auffret, A.G. et al. More warm-adapted species in soil seed banks than in herb layer plant communities across Europe. Journal of Ecology, in press (2023).

Galloway, J.N. et al. Transformation of the nitrogen cycle: recent trends, questions, and potential solutions. Science, 320, 889-892 (2008).

Ghilardi, M. et al. Temperature, species identity and morphological traits predict carbonate excretion and mineralogy in tropical reef fishes. *Nature Communications* 14:985 (2023).

Kambach, S. et al. Climate-trait relationships exhibit strong habitat specificity in plant communities across Europe. *Nature Communications* 14:712 (2023).

LaForgia, M.L. et al. Seed banks of native forbs, but not exotic grasses, increase during extreme drought. *Ecology*, 99, 896-903 (2018).

Sutton, M.A. et al. Challenges in quantifying biosphere – atmosphere exchange of nitrogen species. *Environmental Pollution*, 150, 125-139 (2007).

Reviewer comments, second round review

Reviewer #1 (Remarks to the Author):

Dear Authors,

Thank you very much for the thorough revision. All of my previous questions and comments were addressed in a comprehensive manner, and all issues were clarified sufficiently. I think that the manuscript has further improved in clarity.

I don't have any more questions or comments.

Congratulations for this nice work!

Best Regards,

Orsolya Valkó

Reviewer #2 (Remarks to the Author):

The authors have revised the paper according to the previous comments. I have few minor comments.

Line 79, delete "species".

Line 87, "replacement" should be "input".

Line 196-198, this sentence should be in Methods.

Lines 308-310, need a transition between the two sentences.

Reviewer #3 (Remarks to the Author):

I am very happy with the changes made to the paper and consider it ready for publication.

Lindsay Turnbull

Responses to reviewer comments

REVIEWERS' COMMENTS

Reviewer #1 (Remarks to the Author):

Dear Authors,

Thank you very much for the thorough revision. All of my previous questions and comments were addressed in a comprehensive manner, and all issues were clarified sufficiently. I think that the manuscript has further improved in clarity.

I don't have any more questions or comments.

Congratulations for this nice work!

Best Regards,

Orsolya Valkó

Our response: We are grateful for the excellent and helpful review of our manuscript and these kind words!

Reviewer #2 (Remarks to the Author):

The authors have revised the paper according to the previous comments. I have few minor comments.

Line 79, delete “species”.

Our response: We deleted this word.

Line 87, “replacement” should be “input”.

Our response: We changed this as suggested.

Line 196-198, this sentence should be in Methods.

Our response: As suggested by the editorial team, we connected this sentence better to the following sentence, please see lines 210-211.

Lines 308-310, need a transition between the two sentences.

Our response: We modified this sentence as suggested, please see lines 320-322.

Reviewer #3 (Remarks to the Author):

I am very happy with the changes made to the paper and consider it ready for publication.

Lindsay Turnbull

Our response: Thank you very much, we really appreciate thorough review of our manuscript

that helped to considerably improve it!